# Understanding the Gap between Environmental Intention and Pro-Environmental Behavior towards the Waste Sorting and Management Policy of China

**DOI:** 10.3390/ijerph18020757

**Published:** 2021-01-17

**Authors:** Huilin Wang, Aweewan Mangmeechai

**Affiliations:** International College, National Institute of Development Administration, 118 Moo 3, Sereethai Road, Klong-Chan, Bangkapi, Bangkok 10240, Thailand; huilin.wan@stu.nida.ac.th

**Keywords:** behavior-intention gap, environmental intention, pro-environmental behavior, the theory of planned behavior

## Abstract

Environmental pollution and ecological damage caused by human activities have attracted widespread attention in recent years, and while citizens’ environmental awareness and intentions have increased, their actions may not necessarily change accordingly. This study aims to understand the intention–behavior gap, based on the theory of planned behavior (TPB), by exploring the relationship between intention and pro-environmental behavior on the new waste sorting policy in China. The structural model of extended TPB was tested using sample data from 3113 residents of Changsha, each of whom was asked to complete a two-stage survey. Results demonstrated that perceived policy effectiveness and actual behavioral control positively affect behavioral intention, implementation intention, and pro-environmental behavior. Among them, the actual behavioral control of residents was found to be the most influential factor on behavioral intention and implementation intention, followed by residents’ perceived policy effectiveness. Moreover, behavioral intention and implementation intention mediate the relationship between antecedents and pro-environmental behavior. These findings imply that people with high-level perceptions of policy effectiveness, strong control over actual behavior, strong behavioral intentions, and strong implementation intentions are more likely to engage in pro-environmental behavior. The findings suggest that factors such as perceived policy effectiveness and actual behavioral control should be considered when implementing new policies and campaigns for waste sorting and management.

## 1. Introduction

China has one of the heaviest waste burdens in the world [1]. The total municipal solid waste (MSW) production in 200 large and medium-sized cities in China reached 211,473 million tons in 2018 [2]. The annual growth rate of MSW has remained at 8–10% in the past years [3]. More noteworthy is that due to the massive construction of landfill plants around its cities, two-thirds of China’s large and medium-sized cities are surrounded by garbage [4]. The rapid development of the economy, the acceleration of urbanization, and the growth of the urban population have led to a sharp increase in the generation of MSW [5]. Considerable natural resources are used to produce goods that are consumed and eventually discarded [6]. The uncontrolled growth of MSW has become a potential threat to environmental, social, public health, and economic development [7]. Worryingly, the predicament that China is now facing is also an issue that many other developing countries cannot avoid during the development process. Behind the material prosperity, the environmental pollution caused by waste is self-evident.

Consequently, the Ministry of Housing and Urban-Rural Development of the People’s Republic of China [8] made clear and strict instructions on waste sorting in June 2019. According to the policy, 46 cities should complete the new waste sorting system by the end of 2020 and all cities at the prefecture-level by the end of 2025 [8]. Since 2019, most Chinese cities have successively introduced local mandatory waste sorting policies under the direction of the State Council. Residents’ behavior is the result of the interaction of individual psychological factors and policy interventions. A better understanding of residents’ behavior will help policymakers improve the effectiveness of waste sorting and management policies.

Most research focusing on waste separation and recycling tends to focus on the antecedents that affect residents’ intention and behavior while ignoring the existence of the intention-behavior gap. However, intentions usually only explain 20% to 30% of the variance in behavior [9]. In reality, people find it difficult to get rid of the gap between their good intentions and actual behavior [10]. Even if an individual has a strong intention, they cannot fully guarantee the outcomes that they desire [11]. Zhang et al. [12] confirmed the gap between Chinese people’s behavioral intentions and behaviors in waste recycling, in that many participants with a strong willingness to recycle have yet to act. As Liao et al. [13] suggest, follow-up studies should use more methods to measure actual behavior to investigate the effectiveness of policies. Given the discussion above, this study’s objectives are: (1) to understand the pro-environmental behaviors of Changsha residents under the influence of waste sorting and management policy, (2) to analyze the factors that affect residents’ pro-environmental intentions and behaviors and bridge the intention-behavior gap, and (3) to provide recommendations for related agencies (e.g., local government, policymaker, local community).

This study investigated different behaviors and influencing factors in waste sorting and recycling. However, unlike previous studies that only focus on the antecedents that affect behavioral intentions, this study added factors such as perceived policy effectiveness, actual behavioral control, and implementation intentions to create an extended TPB model, and used a longitudinal research method to demonstrate the whole process from the generation of the behavioral intention to the actual behavior. In this case, Changsha City was selected as the study area. Results showed that residents’ perceived policy effectiveness and actual behavioral control are important factors that affect their behavioral intentions, implementation intentions, and pro-environmental behaviors. This study provides rich information for the development of China’s waste sorting policy through an in-depth analysis of psychological factors in the Chinese context, thereby reducing the government’s environmental pressure and financial burden. The results of this study also promote multidisciplinary applications, such as environmental behavior, behavioral psychology, and public policy.

The rest of the study is structured as follows: Section 2 reviews the literature related to the theory of planned behavior and waste sorting management policies. Section 3 presents the hypotheses and the conceptual model of this study. Section 4 describes the process and method of data collection. Section 5 provides the results of data analysis and hypothesis testing. Section 6 analyzes and discusses the results. Section 7 summarizes the paper, explains the limitations of the study, and provides directions for future research.

## 2. Literature Review

### 2.1. Theory of Planned Behavior

Pro-environmental behavior means that individuals consciously minimize the negative impacts of individual activities on nature, including minimizing resource consumption, reducing waste generation, and recycling [14]. The theory of planned behavior (TPB) has been widely used in recent years to study pro-environmental behavior. Most studies choose TPB as a theoretical model because of its high sufficiency in explaining pro-environmental intention. Most critics accept TPB’s basic behavioral assumptions but question its limitations [15]. Intentions and behaviors are assumed to be the same in many TPB-based studies or are assumed to be highly related [16]. However, Fife-Schaw et al. [17] pointed out that changing the individual’s intention does not guarantee that the individual’s behavior will change accordingly. Scholars call for bridging the intention–behavior gap by considering situational factors (e.g., actual behavioral control, perceived policy effectiveness) [18] and to examine whether individuals have specific plans (i.e., implementation intention) to develop behaviors.

### 2.2. Waste Sorting and Management Policies

The waste management experience of developed countries provides a reference for China. For example, in the United States, command and control instruments and economic incentive instruments are used to a greater [19]. Most states have adopted new regulations to encourage municipalities to provide recycling services to families, and all municipalities are required to develop curbside pricing programs [20]. Economic incentive instruments (e.g., paying for what you throw away) are often used to encourage household waste reduction and recycling in developed countries [21]. Unlike the powerful market forces of MSW management in the United States, the Japanese central government is responsible for providing funding and policy guidance, and local governments at all levels have strong control over regional MSW management as a major executive [22]. In addition, the European Union (EU) has adopted policy instruments on waste management, such as regulations, economic incentives (e.g., landfill taxes), and voluntary measures (e.g., eco-labels), which can fully influence the policies in most of its member states [23].

The goal of the circular economy advocated by the Chinese government in recent years is to maintain the value of resource use and achieve economic development under increasingly severe environmental constraints. The new policy is based on the waste management experience of developed countries and combines China’s basic national conditions and special political and economic systems. The new policy requires residents to divide household waste into four categories, namely recyclable waste, hazardous waste, wet waste (i.e., kitchen waste), and dry waste (i.e., residual waste) [24]. Residents must toss waste into specific public bins at scheduled times when monitors (e.g., community workers, government officials, volunteers) are present to ensure compliance. Local governments have adopted measures combining incentives and punishments. In many cities, residents earn points by correctly classifying household waste. These points can be converted into daily necessities, such as detergents, toilet paper, washing powder, etc. Correspondingly, there are certain penalties for residents and organizations that do not cooperate with this policy. Most of the 46 cities have clarified penalties for illegal discharge of household waste by individuals and organizations, and stipulate that improperly sorted personal waste will be fined 50–200 CNY (7-30 USD) [25]. It is worth mentioning that the measures that combine rewards and punishments have been implemented to varying degrees in cities. In many cities, although local governments have issued relevant regulations and clarified punishment rules [26], relevant law enforcement agencies or departments have not strictly enforced them for various reasons.

## 3. Hypotheses

### 3.1. The Impact of Perceived Policy Effectiveness on Intention and Behavior

Perceived policy effectiveness can be defined as “an individual’s favorable or unfavorable evaluation on the clarity, adequacy, and facilitation of policy measures” [27] (p. 56). When residents have more knowledge about recycling, have a deeper understanding of the environmental benefits of the recycling process, and the greater the actual transparency of the recycling process, the more positive the residents will be to evaluate the effectiveness of the policy. The public’s understanding of the effectiveness of policies and their impact on intentions can provide insights into policy development [28]. When participants believe that policies are not effective in solving collective dilemmas, they are less likely to cooperate and take supportive actions [29]. However, if people are aware of environmental policy issues and the feedback generated by interaction with the environment, the likelihood of their sustainable behavior may increase significantly [30]. In other words, when an individual perceives the existence of more effective policy measures, it is bound to motivate the individual to perform a specific behavior with a higher level of intention. Further, Wan et al. [28] collected survey responses from 198 Hong Kong residents, verifying that perceived policy effectiveness has a significant positive impact on behavioral intentions and contributes 19.39% to explaining recycling intentions. More recently, according to the calculation results of the Garson formula, perceived policy effectiveness as a situational factor has a higher sensitivity to intentions [31]. This study thus proposes the following hypotheses:

**Hypotheses** **1 (H1).**
*Perceived policy effectiveness of urban residents is positively related to their implementation intention.*


**Hypotheses** **2 (H2).**
*Perceived policy effectiveness of urban residents is positively related to their pro-environmental behavior.*


### 3.2. The Impact of Actual Behavioral Control on Behavior

In TPB, perceived behavioral control (PBC) is often used as a proxy for actual behavioral control [32]. Effective behavior prediction requires that PBC accurately reflect actual control, but sometimes PBC cannot do this exactly [33]. Therefore, the gap between an individual’s PBC and actual behavioral control may be hidden under the intention-behavior gap [34]. PBC focuses on the perception or imagination of control, not the actual condition [35]. The importance of actual behavioral control manifests in the fact that the resources and opportunities available (e.g., time, money, skills, the cooperation of others) to the individual determine their behavior to a certain extent [36]. If an individual has the required non-motivational factors mentioned above and intends to perform the act, they will successfully put them into action [32]. Many studies have confirmed that individuals can achieve their intentions, as long as they have sufficient actual control over the behavior [32]. This study thus proposes the third hypothesis:

**Hypotheses** **3 (H3).**
*Actual behavioral control of urban residents is positively related to their implementation intention.*


### 3.3. Mediating Effects of Implementation Intention

When individuals have a very clear vision on when, where, and how they are going to perform their actions, it means they have strong implementation intention [37]. Unlike general intention (e.g., “I intend to separate household waste”), implementation intention requires individuals to imagine how to perform goal-oriented behavior in specific situations. In the case of waste sorting activities, the implementation intention could be expressed as “I intend to separate the household waste and put them into the right garbage bin after I get home from work every day”. Here, opportunities for action (e.g., “after I get home from work every day”) are linked to specific goal-oriented actions (e.g., “separating waste and putting them into the right trash can”), resulting in implementation intention. 

Individuals who have an explicit plan are more committed to choosing action plans and have a higher level of implementation intention [38]. Meanwhile, these people are perceptually ready to encounter situational cues, and these cues can evoke a specific reaction without consciousness [39]. Implementation intention was proven to be effective in changing habitual behavior or promoting new behavior. Carrington et al. [34] developed a model that specifically focused on the gap between ethical purchase intention and actual buying behavior. In their study, intention leads to the formation of implementation intention, which in turn leads to behavior. Therefore, the implementation intention actively mediates the relationship between intention and behavior because these plans can protect their intentions from unnecessary influence. Subsequently, Grimmer and Miles [40] tested Carrington et al.’s [34] model for the consumer’s pro-environment buying behavior, and the results supported their research. This study thus proposes the fourth hypothesis:

**Hypotheses** **4 (H4).**
*Implementation intention mediates the effect of behavioral intention on pro-environmental behavior.*


### 3.4. Mediating Effects of Behavioral Intention

According to TPB, behavioral intention is considered the immediate antecedent of behavior [32]. This statement has been confirmed by many studies, but certain limitations remain. The existence of implementation intentions can greatly improve the realization of goal intentions that are difficult for individuals to achieve [41]. The effectiveness of implementation intentions on behavior has been proven over the years. In a study by Sheeran and Orbell [9], the individual who formed an implementation intention could take vitamin C pills on time and in volume every day. Also, the formation of implementation intentions may help people reduce dietary fat intake [42], and encourage entrepreneurial behavior [43]. The correct sequence of actions should be that the incentive causes the individual to have a general behavioral intention (e.g., I want to separate and recycle household waste), and then further have specific implementation intentions on how to carry out the behavior, and finally put it into action. This study thus proposes the following hypotheses:

**Hypotheses** **5 (H5).**
*Behavioral intention mediates the effect of perceived policy effectiveness on implementation intention.*


**Hypotheses** **6 (H6).**
*Behavioral intention and implementation intention mediate the effect of perceived policy effectiveness on pro-environmental behavior.*


**Hypotheses** **7 (H7).**
*Behavioral intention mediates the effect of actual behavioral control on implementation intention.*


**Hypotheses** **8 (H8).**
*Behavioral intention and implementation intention mediate the effect of actual behavioral control on pro-environmental behavior.*


The conceptual model as shown in Figure 1.

## 4. Methodology

### 4.1. Sampling

Changsha, the capital of Hunan Province, is one of the relatively developed cities in South-Central China and can represent the state of waste management in most Chinese cities to some extent. To investigate the pro-environmental behaviors of Changsha residents, this study conducted a questionnaire survey from May to June 2020. The researcher sought cooperation from community leaders and NGOs in selected municipal districts and asked them to distribute questionnaires to community residents. Moreover, the researchers also contacted government agencies, schools, and companies and invited them to participate in the survey. The above strategies helped researchers to collect a large number of questionnaires in a relatively short period of time. The cluster sampling method was employed in this study. The advantage is that as a random sampling method, it can ensure that all respondents have an equal chance to be selected to participate in this survey (Table 1). The questionnaire was distributed to residents in six districts of Changsha City by mail or in-person. The survey was conducted in two stages. The first stage of sampling continued until about 5,000 initial samples were received. For the first stage, parts 1 to 4 of the questionnaire were distributed (see Section 4.2). One week after the end of the first stage of the survey, respondents were again invited to complete the second stage of the survey, in which parts 5 and 6 of the questionnaire were distributed. After deleting the invalid questionnaire with incomplete information, 3113 samples were selected for further analysis.

### 4.2. The Questionnaire

The questionnaire survey included six parts. The first part was used to collect information about social demographics. The contents of the second part to the sixth part were shown in Table 2. The second part collected relevant data of residents’ perception of waste policy, the scale was developed by Wan et al. [27] and Wan et al. [28]. The third part collected data on residents’ general intentions regarding waste separation and recycling, the measurement derived from Cheung et al. [45] and Terry et al. [46]. The fourth part further collected data about more specific implementation intentions, which were measured using the scale derived from Gollwitzer and Brandstätter [47] and Dholakia et al. [38]. The fifth part collected the data of residents’ actual behavioral control in the past week, which was measured using the scale developed by Rosenthal [30]. The sixth part collected the data of residents’ pro-environmental behavior in the past week, the scale was developed by Cleveland et al. [48]. All the above items were measured using a five-point Likert scale (i.e., 1 = strongly disagree, 5 = strongly agree, or 1 = never, 5 = always).

### 4.3. Data Analysis

In this study, a structural equation model (SEM) with AMOS 23.0 was used to test the hypothetical model. Maximum likelihood (ML) estimation is widely used to analyze most confirmatory factor analysis (CFA) models and is applicable when the measured variables follow a multivariate normal distribution in the population [49]. The absolute value of skew was within 3, and the absolute value of kurtosis was within 7, which was in line with the recommended value of Kline [50], so this study also adopted the ML estimation. In addition, Anderson and Gerbing [51] suggested using a two-step modeling approach in theoretical testing; that is, using SEM to evaluate the measurement model and the structural model. The first stage fully evaluated the construct validity of the model, and the second stage measured the fitting coefficients and path coefficients of the hypothetical model.

## 5. Results

### 5.1. Respondents’ Socio-Economic Characteristics and Pro-Environmental Behavior

The respondents’ socioeconomic background was summarized in Table 3. In terms of age composition, the distribution of all age groups was relatively even. Among them, 47.9% were males and 52.1% were females. In terms of educational level, 33.6% of residents had a bachelor’s degree. In this sample, 28.4% worked in the public sector, 25.5% worked in the private sector, 2.3% were self-employed, and 43.8% were not working. The reason for the high proportion of people who were not working is that this group includes students, retirees, and housewives. Considering that 41.8% of the respondents are 28 years old or younger, it is sufficient to explain the reasonableness of this ratio. A whopping 40.6% of residents said that their monthly income was less than 3000 CNY (430 USD). Also considering the ratio of respondents who were not working, it was sufficient to show that this ratio was reasonable. Comparing the main data from the latest official census report of Changsha, we can see that respondents’ socio-economic background (e.g., age, gender, occupation, monthly salary) matched the wider population of Changsha and was evenly distributed in all corners of Changsha. 

Table 2 showed the results of the pro-environmental behavior survey. More than 50% of respondents indicated that they often or always sort waste in the past week. Approximately 30% of respondents have done it sometimes. More than 10% of respondents admitted that they had rarely or never participated in waste sorting. The findings were consistent with the findings of Xiao et al. [53] in Xiamen, China: 53.5% were always, 37.9% were occasionally, and 8.6% were never.

### 5.2. Measurement Model

Reliability analysis involves testing the Cronbach’α coefficient and the composite reliability (CR) coefficient of the latent variables [54]. The reliability test results in Table 4 showed that the Cronbach’α coefficients of all variables were in the range of 0.896 to 0.940, which indicates consistency among the items of each construct. Similarly, the composite reliability coefficients ranged from 0.898 to 0.940, indicating that items can represent each construct. Convergent validity mainly measures the factor loadings and the average variance extracted (AVE) [54]. All standardized loadings varied from 0.777 to 0.937, which was greater than the critical value of 0.7 suggested by Fornell and Larcker [54]. The lowest value of AVE of all variables was 0.688, which was greater than the benchmark of 0.5. The discriminative validity mainly verifies the relationship between the correlation coefficient between each latent variable and the square root of AVE. As demonstrated in Table 5, the square root value of AVE for all variables was greater than the correlation coefficient between variables. Thus, each variable has good discriminative validity.

### 5.3. Common Method Variance

The potential problem that common method variance (CMV) may exist in behavior research was tested. First, the results of Harman’s single-factor test showed that the percentage variance extracted from a single factor was 53.823% (higher than the classic threshold of 50%), implying the presence of CMV [55]. Second, this study followed the methods suggested by Lindell and Whitney [56] to conduct a CFA factor test. The substantively explained variance (i.e., R1^2^) was 0.256, the average method-based variance (i.e., R2^2^) was 0.516, and the ratio between them was about 1:2. So, this method also proved CMV’s existence. Next, this study applied the unmeasured latent method construct (ULMC) to estimate whether CMV would affect the model [57]. The result showed that the *p*-value of the nested model comparison was 0.987 (*p* > 0.050). Therefore, there is no significant difference, indicating that CMV does not affect the model, and no CMV correction is required in this study.

### 5.4. Structural Model

The results of structural modeling show that the model can be identified and converged, and no negative error variance in the model graph of the non-standardized estimates, indicating that the model identification rules have not been violated. Moreover, it is assumed that the model fits the data well (χ^2^ = 1084.219, df = 96, χ^2^/df = 11.294, CFI = 0.978, GFI = 0.958, AGFI = 0.940, RMSEA = 0.058), and the above values are in line with the fit index value recommended by Hair et al. [58]. Although the ratio of chi-square and degree of freedom is too large (χ^2^/df > 3), according to Hair et al.’s [58] suggestion, when the number of samples is too large, the chi-square value should not be used as the appropriate index value for the model fit testing.

Baron and Kenny [59] suggested that the existence of the mediating effect must meet the following three conditions: (a) the independent variable significantly affects the mediator; (b) the mediator must significantly affect the dependent variable; and (c) the independent variable significantly affects the dependent variable. Hence, this study followed the causal method to test the first mediation condition regarding hypotheses 1, 2, and 3. As shown in Table 5, the correlation coefficients showed that perceived policy effectiveness was positively and significantly related to implementation intention (r = 0.611, *p* < 0.01), and was also positively and significantly related to pro-environmental behavior (r = 0.494, *p* < 0.01). Moreover, actual behavioral control was positively and significantly related to implementation intention (r = 0.708, *p* < 0.01). In addition, as shown in Figure 2, the results of the direct effect of perceived policy effectiveness on implementation intentions (standardized direct effect = 0.28, *p* < 0.001), the direct effect of perceived policy effectiveness on pro-environmental behavior (standardized direct effect = 0.58, *p* < 0.001), and the direct effect of actual behavioral control on implementation intentions (standardized direct effect = 0.61, *p* < 0.001) were statistically significant. Therefore, hypotheses 1, 2, and 3 were supported.

This study further verified Hypothesis 4 to Hypothesis 8 measuring the second condition of mediation. As shown in Table 5, the correlation coefficients indicated that perceived policy effectiveness was positively and significantly related to behavioral intention (r = 0.589, *p* < 0.01), actual behavioral control was positively and significantly associated with behavioral intention (r = .636, *p* < 0.01), behavioral intention was positively and significantly associated with implementation intention (r = 0.799, *p* < 0.01), and implementation intention was positively and significantly associated with pro-environmental behavior (r = 0.570, *p* < 0.01). Moreover, Figure 3 added mediators on the basis of Figure 2, thus showing the results of the direct effects of perceived policy effectiveness on behavioral intention (standardized direct effect = 0.33, *p* < 0.001, see Figure 3), the direct effect of actual behavioral control on behavioral intention (standardized direct effect = 0.47, *p* < 0.001), the direct effect of behavioral intention on implementation intention (standardized direct effect = 0.57, *p* < 0.001), and the direct effect of implementation intention on pro-environmental behavior (standardized direct effect = 0.47, *p* < 0.001) were all statistically significant. The second condition of mediation was therefore supported.

Although the Sobel test is usually used for mediation analysis, it may not be suitable for testing the importance of mediation effects in some cases [60]. Bollen and Stine [61] proposed to use the bootstrapping approach to improve the accuracy of indirect influence confidence when verifying the mediation effect. This method allows multiple mediators to be tested simultaneously and is considered suitable for this study. Consequently, this study followed the suggestions of Preacher and Hayes [62]: 5000 bootstrap samples were generated with bias-corrected and percentile bootstrapping at a 95% confidence interval. If a value of zero does not appear within the 95% confidence interval, the effect is considered significant. The standardized results of the bootstrapping approach are presented in Table 6. The Z values were all greater than 1.96, and no value of zero was found within the 95% confidence interval. Therefore, this study confirmed the existence of positive and significant mediating effects for behavioral intentions between perceived policy effectiveness and implementation intentions (standardized indirect effect = 0.187, *p* < 0.001), and Hypotheses 4 was thus supported. This finding indicated that people with high-level perceptions of policy effectiveness and strong behavioral intentions are more likely to have strong implementation intentions. Moreover, it appears that the positive and significant mediating effects for implementation intentions between behavioral intentions and pro-environmental behavior (standardized indirect effect = 0.272, *p* < 0.001), and Hypotheses 5 was thus supported. This showed that people with strong behavioral intentions and strong implementation intentions are more likely to engage in pro-environmental behavior. Furthermore, there were positive and significant mediating effects for behavioral intentions and implementation intentions between perceived policy effectiveness and pro-environmental behavior (standardized indirect effect = 0.120, *p* < 0.001), and Hypotheses 6 was thus supported. This means that people with high-level perceptions of policy effectiveness, strong behavioral intentions, and strong implementation intentions are more likely to engage in pro-environmental behavior. In terms of the actual behavioral control, this study figured out that the positive and significant mediating effects for behavioral intentions between actual behavioral control and implementation intentions (standardized indirect effect = 0.271, *p* < 0.001), and hypothesis 7 was thus supported. This suggested that people with strong control over actual behavior and strong behavioral intentions are more likely to have strong implementation intentions. Also, there were positive and significant mediating effects for behavioral intentions and implementation intentions between actual behavioral control and pro-environmental behavior (standardized indirect effect = 0.293, *p* < 0.001). The findings imply that people with strong control over actual behavior, strong behavioral intentions, and strong implementation intentions are more likely to engage in pro-environmental behavior. 

## 6. Discussion

### 6.1. Contributions

First, this study developed a new model based on TPB. As displayed in Figure 3, data analysis results show that 42% of the variance in behavior can be explained in this model. This is the first study to consider actual behavioral control and perceived policy effectiveness to bridge the intention-behavior gap. Especially in China, the implementation of a compulsory waste sorting policy is in its infancy, and residents may have different levels of understanding of the policy. Policymakers find it difficult to grasp the prompt implementation of the policy at the grass-roots level. Residents act as policy implementers, so their attitudes and reactions will inevitably affect policy implementation. More specifically, policymakers must understand the relationship between policy measures and behavior from the perspective of citizens, which will inspire sustainable policy design and formulation. Therefore, waste management is not only a matter for government agencies, but also must hold all citizens and stakeholders accountable for their actions.

Second, this study’s findings generally indicated that perceived policy effectiveness and actual behavioral control were both related to pro-environmental intentions. Increasing residents’ perception of policies’ effectiveness can enhance their pro-environmental intentions and behaviors. Also, if residents feel the increased control over their behavior in actual situations, they will form more specific implementation intentions. Compared with perceived policy effectiveness, actual behavioral control has a greater impact on implementation intention. Furthermore, the results revealed that behavioral intention and implementation intention mediated the relationship between antecedents (i.e., perceived policy effectiveness, actual behavioral control) and pro-environmental behavior. This result was in line with Gollwitzer’s [63] research on implementation intention as a mediating role.

### 6.2. Implications

Perceived policy effectiveness has a significant positive impact on behavioral intention, implementation intention, and pro-environmental behavior. Among them, perceived policy effectiveness has the strongest impact on behavioral intention, followed by its influence on pro-environmental behaviors, and its influence on implementation intentions is the weakest. To improve residents’ behavioral intentions and encourage pro-environmental behaviors, the government should consider how to increase residents’ perception of the effectiveness of policies. For example, the government can conduct systematic school environmental education and incorporate environmental education courses into the credit evaluation program to improve the recycling knowledge of the student population. For another example, the government can place public service advertisements on TV programs and websites to enable residents to understand the environmental benefits of recycling. However, the implementation of policies is considered to be the weakest link in environmental protection. In many cases, the government has launched good policies, but without effective publicity and long-term supervision, the policies are ultimately just regulations written on paper and have not exerted social effects. Taking the waste sorting in Changsha as an example, the researchers visited some communities and communicated face-to-face with residents. Every community can see signs and campaigns about waste sorting and management policy. However, many residents who lack consciousness can simply just neglect the regulations.

During the first phase of implementation, the enthusiasm of residents actively involved in waste separation and recycling may be weakened. The government can apply a rewarding system to these residents appropriately. In the meantime, imposing fines for irresponsible actions is another effective method. When people need to pay a price for their irresponsible environmental behavior, it will force them to make adjustments and behave responsibly. The punishment should be reasonable and combined with the government’s publicity and education methods. Moreover, a serious monitoring system e.g., designated staff to provide guidance and supervision around the trash area is necessary.

Actual behavioral control also has a significant positive impact on behavioral intention and implementation intention. In order to help residents, develop pro-environmental intentions, especially their implementation intentions, it is essential to improve actual behavioral control. Actual behavioral control includes knowledge/skills [64,65] and public facilities. On the one hand, the government should undertake effective policy publicity and education to ensure that every resident has the skills to separate waste. Public environmental education should not only provide information to the public but also support hands-on experience in waste sorting. For example, explaining to the public why waste is divided into four categories (i.e., dry waste, wet waste, recyclable waste, hazardous waste) is more effective than directly asking the public to sort waste. On the other hand, the government should prepare enough facilities for waste sorting practice because convenience is one of the key factors in pro-environmental behavior. The government should regularly check whether there are enough garbage bins in the community and whether the waste collection and transportation facilities meet the standards. Moreover, while further promoting waste sorting policies, the government should think about how to improve the acceptability of regulations. The awakening of consciousness has never been achieved overnight and usually requires long-term monitoring of government policies and investment in infrastructure.

## 7. Conclusions

Based on the proposed objectives, this study investigated the pro-environmental behavior of Changsha residents under the guidance of the waste sorting and management policy, and explored the relationship between perceived policy effectiveness, actual behavioral control, and pro-environmental behavior to bridge the intention–behavior gap. The findings prove that high-level pro-environmental behaviors are more likely to appear in residents with a high degree of control over actual behavior, those with a strong perception of effective policies, and those with a strong implementation intention. This study focuses on the intention-behavior gap and adopts a longitudinal research approach (i.e., a two-stage survey) to examine the behavioral mechanisms of individuals. This breaks the situation where previous research mostly stops at behavioral intentions and provides a reference for the subsequent research of pro-environmental behavior to shift from cross-sectional research to longitudinal research. Moreover, the research conclusions provide a basis for government agencies, communities, and schools to further promote waste sorting and management policies. It is recommended that the government strengthen supervision during the implementation of waste sorting and management policy, especially at the community and school levels. It is also worth mentioning that this study is based on China’s special policy environment and cultural background. The government has adopted a combination of different policy instruments to implement policies. However, in the early stages of policy implementation, command-and-control policy instruments were widely used. Therefore, whether the research results can be consistent with the research of other countries remains to be verified, especially for countries with relatively mature waste management systems (e.g., Germany, Japan) and countries with economic incentive instruments as their main policy instruments (e.g., EU countries).

This study has certain limitations. First, the data in this study were collected from residents’ self-reports, but self-reported behavior may deviate from actual behavior. Inevitably, some residents may beautify their behavior to meet the needs of social morality. This leads to an increased probability of CMV. Follow-up studies should prevent potential CMV in advance. That is, researchers should carefully design questionnaires, use tailor-made CMV measures, and use archive data or contact multiple respondents. However, if CMV exists, it can be checked and corrected afterward. Second, this study did not consider the important factor of individual habits. However, implementation intentions sometimes provide additional advantages for individuals whose status or habits affect the progress of achieving the goal. It is not easy for individuals with fixed behavior patterns to change their implementation intentions. Therefore, individual habits should be taken into consideration in the subsequent research on the intention–behavior gap. Third, this study did not consider the moderating effect of perceived policy effectiveness on the relationship between intention and actual behavior. Follow-up studies can try to explore this direction, that is, to compare the moderating effect of different types of policy instruments (i.e., command and control, economic incentive, voluntary) on the relationship between intention and actual behavior.

## Figures and Tables

**Figure 1 ijerph-18-00757-f001:**
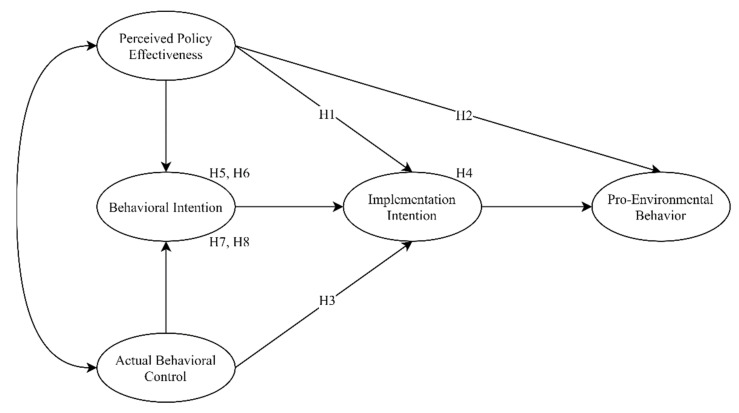
Conceptual model.

**Figure 2 ijerph-18-00757-f002:**
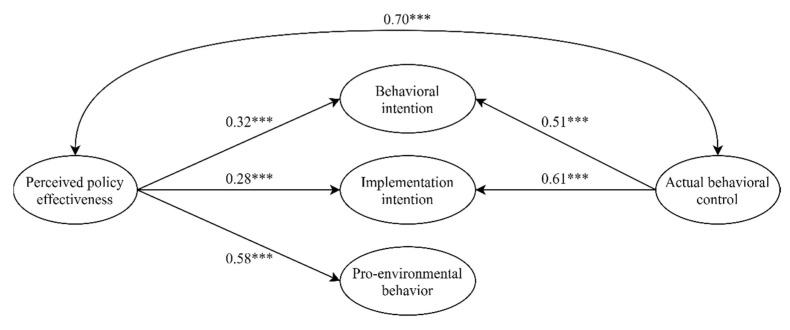
Direct effects of perceived policy effectiveness and actual behavioral control on behavioral intentions, implementation intention and pro-environmental behavior. *** *p* < 0.001; Standardized coefficients are reported.

**Figure 3 ijerph-18-00757-f003:**
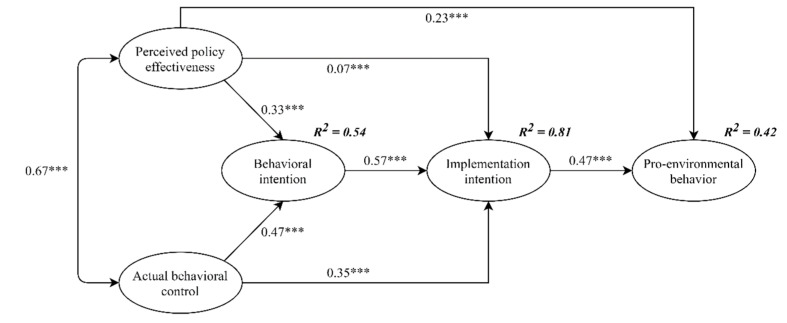
Structural equation modeling of the hypothesized model. *** *p* < 0.001; Standardized coefficients are reported.

**Table 1 ijerph-18-00757-t001:** Number of questionnaires distributed and recovered in municipal districts.

Municipal Districts	Population (One Hundred Thousand) ^1^	Proportion (%)	Number of Questionnaires Distribution	Number of Respondents
Furong	5.954	14	700	433
Tianxin	6.643	15	750	495
Yuelu	8.496	20	1000	640
Kaifu	6.369	15	750	465
Yuhua	8.880	21	1050	618
Wangcheng	6.333	15	750	462
Total	42.675	100	5000	3113

^1^ Hunan Provincial Bureau of Statistics [44].

**Table 2 ijerph-18-00757-t002:** Pro-environmental behavior and frequency (N = 3113).

Variables	Categories (%)				
	Never	Rarely	Sometimes	Often	Always
How often do you separate wet waste?	1.9%	10.2%	35.0%	36.9%	15.9%
How often do you separate dry waste?	1.8%	8.9%	31.4%	41.4%	16.6%
How often do you separate waste for recycling purposes?	1.8%	9.8%	30.5%	40.0%	17.9%

**Table 3 ijerph-18-00757-t003:** Characteristics of Changsha respondents (N = 3113).

Parameter	Characteristics		This Study (Percent)	2017 Census Estimates ^1^
Age	≤17		22.1	≤14 (13.6%)
	18–28		20.7	15–64 (77.4%)
	29–44		30.9	-
	45–59		20.2	-
	≥60		6.1	≥65 (9.0%)
Gender	Male		47.9	50.0%
	Female		52.1	50.0%
Education level	Below high school		16.1	
	High school or vocational certificate		13.0	
	Higher vocational certificate		28.5	
	Bachelor’s degree		33.6	
	Master’s degree or above		8.8	
Occupation	Public sector		28.4	24.5%
	Private sector		25.5	23.1%
	Self-employed		2.3	21.0%
	Unemployed (e.g., housewives, students, retirees, etc.)		43.8	-
Monthly salary (CNY)	≤3000		40.6	Mean 7628 CNY
	3001–5000		18.4	
	5001–10,000		27.3	
	10,001–20,000		11.5	
	≥20,001		2.2	

^1^ Changsha Municipal Bureau of Statistics [52].

**Table 4 ijerph-18-00757-t004:** Results of confirmatory factor analysis.

Construct	Items		Loadings	Cronbach’s α	CR	AVE
PPE	PPE1	The environmental programs organized by the government effectively arouse the environmental awareness of the general public.	0.777	0.896	0.898	0.688
	PPE2	The government provides clear guidelines and regulations on household waste separation and recycling.	0.886			
	PPE3	The government promotes clearly household waste separation and recycling as positive symbols, labels, images, and events.	0.855			
	PPE4	Overall, the compulsory policy on waste sorting and recycling implemented by the government is effective.	0.795			
ABC	ABC1	During the previous week, I was able to separate and recycle household waste as I planned.	0.872	0.899	0.899	0.748
	ABC2	During the previous week, I was able to find separate garbage bins.	0.840			
	ABC3	During the previous week, I was certain and confident about my separation and recycle knowledge/skills.	0.882			
BI	BI1	I will separate and recycle household waste.	0.908	0.940	0.940	0.839
	BI2	I want to separate and recycle household waste.	0.916			
	BI3	I intend to engage in household waste separation and recycling.	0.924			
IMP	IMP1	For the next garbage discard, I plan to separate them in advance when I have free time (e.g., before and after work, before and after school).	0.854	0.922	0.923	0.799
	IMP2	For the next garbage discard, I plan to put paper waste and plastic bottles into recycle trash bin provided by the Government.	0.905			
	IMP3	For the next garbage discard, I plan to put wet waste (i.e., kitchen waste, food waste) into the designated trash can within the stipulated time.	0.922			
PEB	PEB1	During the previous week, how often do you separate wet waste?	0.913	0.921	0.923	0.800
	PEB2	During the previous week, how often do you separate dry waste?	0.937			
	PEB3	During the previous week, how often do you separate waste for recycling purposes?	0.830			

**Table 5 ijerph-18-00757-t005:** Means, standard deviations, reliabilities, and correlations.

Construct	Mean	S.D.	PPE	ABC	BI	IMP	PEB
PPE	4.029	0.652	**(0.829)**				
ABC	3.906	0.698	0.602 **	**(0.865)**			
BI	4.194	0.601	0.589 **	0.636 **	**(0.916)**		
IMP	4.113	0.614	0.611 **	0.708 **	0.799 **	**(0.894)**	
PEB	3.597	0.871	0.494 **	0.678 **	0.504 **	0.570 **	**(0.894)**

The square root of AVE is in diagonals (bold), and off diagonals are Pearson correlation of constructs. ** *p* < 0.01.

**Table 6 ijerph-18-00757-t006:** Standardized direct, indirect, and total effects.

	Point Estimate	Product of Coefficients		Bootstrapping				
				Percentile 95% CI		Bias-Corrected 95% CI		Two-Tailed Significance
		*SE*	*Z*	Lower	Upper	Lower	Upper	
*Direct effects*								
PPE → BI	0.325	0.028	11.607	0.271	0.380	0.268	0.378	0.001 (***)
PPE → IMP	0.066	0.021	3.143	0.025	0.107	0.024	0.106	0.002 (**)
PPE → PEB	0.229	0.029	7.897	0.170	0.283	0.169	0.281	0.001 (***)
ABC → BI	0.472	0.027	17.481	0.418	0.524	0.419	0.525	0.000 (***)
ABC → IMP	0.346	0.025	13.840	0.299	0.396	0.299	0.396	0.000 (***)
BI → IMP	0.574	0.023	24.957	0.527	0.620	0.528	0.621	0.000 (***)
IMP → PEB	0.475	0.027	17.593	0.420	0.527	0.422	0.529	0.000 (***)
*Indirect effects*								
PPE → IMP	0.187	0.018	10.389	0.153	0.224	0.153	0.223	0.000 (***)
PPE → PEB	0.120	0.014	8.571	0.094	0.148	0.094	0.148	0.000 (***)
ABC → IMP	0.271	0.019	14.263	0.233	0.309	0.235	0.311	0.000 (***)
ABC → PEB	0.293	0.024	12.208	0.246	0.340	0.247	0.341	0.000 (***)
BI → PEB	0.272	0.016	17.000	0.241	0.303	0.242	0.305	0.000 (***)
*Total effects*								
PPE → BI	0.325	0.028	11.607	0.271	0.380	0.268	0.378	0.001 (***)
PPE → IMP	0.253	0.027	9.370	0.200	0.305	0.200	0.305	0.000 (***)
PPE → PEB	0.348	0.024	14.500	0.300	0.395	0.299	0.395	0.000 (***)
ABC → BI	0.472	0.027	17.481	0.418	0.524	0.419	0.525	0.000 (***)
ABC → IMP	0.617	0.025	24.680	0.567	0.666	0.568	0.667	0.000 (***)
ABC → PEB	0.293	0.024	12.208	0.246	0.340	0.247	0.341	0.000 (***)
BI → IMP	0.574	0.023	24.957	0.527	0.620	0.528	0.621	0.000 (***)
BI → PEB	0.272	0.016	17.000	0.241	0.303	0.242	0.305	0.000 (***)
IMP → PEB	0.475	0.026	18.269	0.420	0.527	0.422	0.529	0.000 (***)

Standardized estimating of 5000 bootstrap samples, ** *p* < 0.01, *** *p* < 0.001.

## Data Availability

Not applicable.

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
