# Peer review of "Understanding the Gap between Environmental Intention and Pro-Environmental Behavior towards the Waste Sorting and Management Policy of China"

_ijerph, 2021, doi:10.3390/ijerph18020757_

Round 1

Reviewer 1 Report

The paper addresses waste sorting behaviour, taking the example of the Chinese city Changsha. It develops a quantified model based on a conceptional approach differentiating perceived policy effectiveness, intentions and actual behaviour. The model is tested based on empirical data gathered by a questionnaire sent filled in by over 3.000 habitants.

It provides highly interesting insights into different drivers and causal relations between waste sorting policies and individual factors that result in specific environmental outcomes. The clear strength of this paper is the quantitative data analysis with very thorough revisions of data quality and robustness of results. Against this background I would recommend publication in this journal after minor revisions that should refer to the framing of the study and research questions and especially the discussion of results.

The literature review in Chapter 2 should more explicitly refer to international experiences on the introduction of waste sorting policies. As described most cities face similar challenges with regard to motivating inhabitants for proper waste sorting.

In Chapter 3, with regard to the concept of “perceived policy effectiveness”, the authors should discuss in more details how knowledge about recycling processes, understanding of its environmental benefits or the actual transparency of waste flows influence the individual evaluation of policy measures.

In Chapter 4, the extremely high share of inhabitants who responded to the questionnaire should be discussed. Who did send out the questionnaire and were there any possible negative/ positive consequences anticipated by the interviewees?

Chapter 6, line 387: It doesn´t seem very surprising that higher fees for neglecting sorting policies would lead to improved sorting qualities. Please discuss if e.g. such economic incentives would modulate intentions or rather reduce the gap between intention and actual behaviour.

The conclusions should also include a discussion about the specific cultural context of Chinese environmental policies: Would similar results to be expected e.g. in European countries?

Reviewer 2 Report

18/ Second half of the abstract is not clear enough and had to be read multiple times in order to get the idea about the content of the paper – simplify it or make it somehow clearer to the reader.

58/ Objectives of the paper are not stated well – to understand the problem? to understand the gap? These are not relevant scientific goals. Reformulate them into proper objectives/goals.

70/ Literature review should be significantly extended, provide additional comparison with measures used in other countries, differences between types of rewardsm incentices, etc. – the theory you are building your paper on.

84/ Stress more that you are describing China WM policy, elaborate more on how in/proper behavior is checked by the authorities, resp. overall how the incetives/punishment system is set up and how it works in practice.

165/ I tend to get lost in so many hypotheses that seem stated in sort of criss-cross manner (it almost seemst like you are examinign effect of A on B and then effect of B on A) → try to make thi part somehow clearer or try to reduce the amount of individual hypotheses or maybe combine them somehow

178/ Cluster sampling chosen because random sampling? Sentence does not make sense. Also elaborate more on how people were selected or who applied to participate in data collection.

226/ Add more comments to what extent collected sample reflects population structure of the analysed city (comment that it "matched" is not enough).

233/ Unfinished sentence?

248/ Under table 4 you not 3 significance level marked by asterisks → but there is not a single asterisk in the whole table → so nothing in table 4 is significant?

330/ Overall, quite large part of the paper is devoted to reporting various statistical values of the analysed sample, but the main idea of the paper seems to get lost among all those data → I would like to see much closer proportion of the "waste behavior" part and the "statistical analysis of the data" part. Because now the result is primarily descriptive, that there is a gap (was expected) and hypotheses hold (also expected from the way hypotheses are defined, plus the observations are to high degree obvious with such hypotheses).

355/ So the outcome of the paper is that supervision should be strenghened? Not very strong outcome. Implementation part of the paper is relatively weak compared to the rest – mostly general thoughts, no reflection of available empirical results of theory. Ideally there should be some evidence or data about how change in some measure or certain factor (that can be affected by municipality or the officials) influences the result. Because no the paper is strong in the data analysis, but the whole point why this data analysis was performed and what is it good for tends to get lost → the benefit should be stated much more and it should also be much more tangible. But this might require significant changes in the way how the whole story about the purpose of this paper and the way how objectives of the paper are defined.

Reviewer 3 Report

The article presented is about something that, a priori, it is very difficult to valorate. The gap between environmental intention and pro-environmental behavior is in fact an important gap in all of the societies. Both necessaries, pass from intention to actions, is sometimes not easy.

The paper is well written and describes precissely all the aspects related to the survey in detail, enough to understand the difficulties and the results. Moreover, authors recognised certain limitations that I expect to be solved in future research.

The behavior of a society, a community is sometimes the sum of the individual behaviors but policy regulate this, promoting a social behavior. I suggest that this would be additional information that can enrich the article or, in a future, the subsequent research.

The introduction, methodology, results and discussion are adequate as well as the conclusions.

First of all, the authors have the following objectives: 1) to understand the problems and challenges faced by Changsha municipality in implementing waste sorting and management policy, 2) to understand the intention-behavior gap in implementing waste sorting and management policy by measuring actual behavioral control and perceived policy effectiveness, and 3) to provide recommendations for related agencies (e.g., local government, policymaker, local community). To measure these objectives, in some way, a questionnaire to selected participants was used based on several questions. Most of them are close to the policy and characteristics of the municipality. So, I think they are in the right way because I have no data about the policy applied in the area. Maybe the first suggestion to add an annex with a summary of the local/governmental policy applied.  

The structure of the text, the confirmatory analysis and the treatment of data seem to be adequate.  

As a major suggestion, I would like to be reflected in the conclusions, the proposed objectives and the answers to these should be, more or less, in the same order in this part of the article.   

Of course, I believe that the main target is to focus on the intention-behavior gap and authors adopt a longitudinal research approach to examine the behavioral mechanisms of individuals. However, how can authors join this with the policy applied? Which are the recommendations for related agencies? Authors include in part of the text some recommendations like this (line 354-357) "Perceived policy effectiveness has a significant positive impact on behavioral intention, implementation intention, and pro-environmental behavior. Therefore, it is recommended that the government strengthen supervision during the implementation of waste sorting and management policy, especially at the community and school levels". It would be useful to indicate clearly the main recommendations in the conclusions. 

Reviewer 4 Report

Dear Authors

The topic of the manuscript “Understanding the Gap between Environmental Intention and Pro-Environmental Behavior towards the Waste Sorting and Management Policy of China” in my opinion will very interesting for readers of this Journal. Authors had a good idea for a research project. In my opinion the research is important to the scientific field. The subject is relevant, the analytical methodologies are adequate, and the volume of data seems to be enough for publication. Methodology is well explained. I have no hesitation in recommending publication following minor revision.

General comments:

Abstract: Abstract presents summary, include key findings and the length of this part of the manuscript is appropriate.

Keywords:

Waste sorting policy and environmental protection - should be deleted or changed

Introduction and Literature Review: I consider that the structure of this section was well designed. Literature Review is adequate. Is effective, clear and well organized.

Some specific comments:

Those two parts can be connected in one …

Waste management and waste sorting is a part of circular economy model and sustainable development also. I thing that it is necessary to add some information about impact of your research on circular economy. I believe that many researchers who are interested in CE, will be interested in your research.  

The aim of the research - please specify in the main text of manuscript (not only hypotheses)

Material and methods: The methodology is well thought through.

Results, discussion and conclusion: The results of the study are well presented.  

In table 2 and 3 – In my opinion the share of respondents is enough to present (the numbers is not necessary)

Discussion is the weakest part of this text. I think that some text from results can be moved to discussion secession. Please consider using of old literature sources …. I think that some of them can be change.

Reviewer 5 Report

Congratulations to the authors for their research, but the following changes are needed to improve it:

  1. In the introduction, I consider it necessary to highlight the connection between the research objectives and the research hypotheses. The novelty of the study and the highlighting of the main conclusions should also be emphasized. A description of the structure of the paper must be added.
  2. Literature Review - can be extended, given that internationally there are a multitude of studies on the Theory of planned behavior, waste sorting and management policies.
  3. It is necessary to add the questionnaire in the Appendix.
  4. Methodology - questionnaire - it necessary to add the way of distributing the questionnaire and the period in which the data were collected.
  5. Line 202 - mentions that the items were analyzed using the Likert scale. I think it can be added Likert scale to the Appendix.
  6. The validation or invalidation of research hypotheses should be added to the results.
  7. From my point of view in the conclusions section the citations should be eliminated (they can be moved in discussions) and the conclusions should express more concisely the research results and also underlined the future research directions.

Round 2

Reviewer 5 Report

The article has been improved. In my opinion it can be published.